# Adolescents’ Experiences of Facilitators for and Barriers to Maintaining Exercise 12 Months after a Group-Based Intervention for Depression

**DOI:** 10.3390/ijerph18105427

**Published:** 2021-05-19

**Authors:** Evelina Sunesson, Emma Haglund, Ann Bremander, Håkan Jarbin, Ingrid Larsson

**Affiliations:** 1Center of Research on Welfare, Health and Sport, School of Health and Welfare, Halmstad University, SE-30118 Halmstad, Sweden; evelina.sunesson@fou-spenshult.se; 2Spenshult Research and Development Centre, SE-30274 Halmstad, Sweden; emma.haglund@hh.se (E.H.); ann.bremander@fou-spenshult.se (A.B.); 3Section of Rheumatology, Department of Clinical Sciences, Lund University, SE-22242 Lund, Sweden; 4The Rydberg Laboratory for Applied Sciences, Halmstad University, SE-30118 Halmstad, Sweden; 5Department of Regional Health Research, University of Southern Denmark, DK-5230 Odense, Denmark; 6Danish Hospital for Rheumatic Diseases, University Hospital of Southern Denmark, DK-6400 Sonderborg, Denmark; 7Child and Adolescent Psychiatry, Department of Clinical Sciences Lund, Lund University, SE-22184 Lund, Sweden; hakan.jarbin@regionhalland.se; 8Child and Adolescent Psychiatry, Region Halland, SE-30185 Halmstad, Sweden

**Keywords:** adolescents, facilitators, barriers, depression, exercise

## Abstract

Exercise can improve health among adolescents with depression. Understanding facilitators for and barriers to maintaining exercise among adolescents with depression may increase adherence to exercise and consequently improve health. The aim was to explore adolescents’ experiences of facilitators for and barriers to maintaining exercise after a group-based exercise intervention for depression. Interviews (*n* = 14) were conducted 12 months after the exercise intervention. A qualitative content analysis was used to extract facilitators and barriers. Facilitators for maintaining exercise among adolescents with depression were (1) greater self-esteem by having companionship while exercising and by achieving exercise results and (2) having a supportive environment in terms of accessibility and coaching. Barriers to maintaining exercise were (1) disease burden due to fatigue, social anxiety, and a lack of drive, and (2) lack of a supportive environment including a lack of social support, as well as structural support. In conclusion, facilitating enhanced self-esteem and continuous support for exercising are important for adolescents with depression to maintain exercise. The disease burden of depression is a substantial barrier that needs to be considered to maintain exercise. The findings can contribute to the development of services that promote and coordinate exercise as a treatment among adolescents with depression.

## 1. Introduction

Depression is one of the leading causes of morbidity, disability, and mortality, globally, among adolescents (10–19 years of age) [1,2,3,4,5], with an increasing prevalence after puberty [6,7]. Depression in adolescence is associated with behaviour disorders, addiction, self-harm, and interrupted schooling, which affects health and quality of life [8,9]. Early treatment might improve health by reducing the persistence or severity and prevent depression relapse later in life [6,9]. Available treatments with pharmacological and cognitive behavioural therapy [10,11] have shown modest efficacy among adolescents [4,12,13,14]. Exercise has been reported as a promising adjunctive treatment in reducing depressive symptoms among adolescents [15,16,17,18]. Adolescents have become more physically inactive in recent decades [19] and adolescents with depression are even more inactive than healthy peers [20,21]. Evidence suggests that the exercise mode should be aerobic, of at least moderate-intensity exercise at 60–70% of maximum heart rate, to reduce depressive symptoms among adolescents and to prevent relapse of depression later in life [22,23,24,25,26]. The most challenging part of an exercise intervention is to maintain the acquired exercise level over time. Exploring facilitators for and barriers to maintaining exercise in adolescents with depressive disorders may help to improve interventions for exercise in clinical practice [27,28,29]. Maintenance of exercise in the general population is dependent on motivational, psychological, supportive, and environmental factors [28,30]. Support from friends and families facilitates maintenance of exercise among healthy adolescents and adults with depression [28,30,31]. Barriers to maintaining exercise among healthy adolescents have been identified as limited financial support from parents and long distance to sports clubs [32]. Negative experiences of exercise have further been found as a barrier to maintaining exercise, where healthy girls expressed feelings of being afraid to embarrass themselves due to lack of skills when performing an exercise [32]. Experiences of stress, depressive symptoms, and low energy have further been identified as barriers to maintain exercise among adults with depression [33], while studies focusing on adolescents with depressive disorders are scarce [34]. 

In summary, exercise is a promising treatment option for reducing depression among adolescents. Difficulties in adhering to exercise routines have been reported, which supports the importance of identifying facilitators for and barriers to maintaining exercise among adolescents with depression. Knowledge of these factors can help the clinic to tailor the treatment. The aim of the present study was to explore adolescents’ experiences of facilitators for and barriers to maintaining exercise after a group-based exercise intervention for depression.

## 2. Materials and Methods

### 2.1. Design

The study had an explorative design where qualitative content analysis was used together with an inductive approach [35,36]. Qualitative content analysis is a research method aimed at systematically analysing qualitative data to provide knowledge, which is shaped within an interaction between the researcher and the participant [35,37]

### 2.2. Participants

This qualitative study was based on a 12-month follow-up of a group-based aerobic exercise intervention aimed at reducing depression in adolescents. Participants were recruited by physicians at the Child and Adolescent Psychiatry (CAP) outpatient clinic in the south-west of Sweden, described in detail elsewhere [38]. Inclusion criteria were: adolescents aged 13–17 years with major depressive disorder [7] and not significantly improved in spite of at least four visits to a CAP-clinic. Exclusion criteria were exercising more than 150 min/week of moderate intensity or 75 min/week vigorous intensity, having eating disorder, intellectual disability or autism precluding participation in groups, physical illness precluding exercise, high suicide risk or chaotic social circumstances interfering with a regular exercise schedule, ongoing psychotherapy or pharmacological therapy adjustment within the past four weeks prior to entering the exercise intervention [38]. Nineteen adolescents with a depression duration between 1.4–5.3 years (median 2.2 years), comorbid with attention deficit hyperactive disorder disorder (71%), and anxiety disorders (62%) participated in a 14-week group-based exercise intervention. The participants received personalised recommendations and were asked to plan an exercise regime at the end of the intervention to maintain their exercising in order to encourage them to continue with exercise after the intervention. Each participant was also offered three follow-up sessions with a licensed personal trainer to support self-management strategies [38]. In the 12-month follow-up study, participants who completed the intervention with a participation of 50% or more (*n* = 16) were invited to a follow-up interview. Fourteen adolescents, ten females and four males aged 14–19 years (median 17 years), accepted the invitation and participated in the interviews. To further describe the characteristics of the participants at the one-year follow-up, body mass index and Quick Inventory of Depressive Symptomatology (QIDS) Adolescent Version [7,39] were used (Table 1). 

### 2.3. Data Collection

Semi-structured follow-up interviews were conducted at the CAP clinic in March-April 2019. The interviews were performed by the researcher (IL) and had a duration of 21–46 min with a median of 36 minutes and a total interview length of 8 h and 9 min. All interviews were initiated with open-ended questions: “Tell me about your exercise routine today and how you experience it”, “What enables you to continue exercising?”, “Which facilitators have you experienced with your ongoing exercise?”, “What barriers have you experienced with your ongoing exercise?”, “What barriers exist for you to continue exercising?” and “How would you like to exercise?”. Follow-up questions such as ‘‘Please, tell me more about...?’’, or ‘‘How do you mean…?” were used to gain depth in the data. A pilot interview was conducted to test the accuracy of the questions and this interview was included in the study because no amendment was required. The interviews were digitally recorded and transcribed verbatim.

### 2.4. Data Analysis

The text was analysed using manifest qualitative content analysis in accordance with Graneheim and Lundman [35]. At first, the main author (ES) listened to and read through the transcribed text several times to understand and gain an overview of the material. Next, text related to the aim was divided by the main author into 440 meaning units and sorted into the two content areas, facilitators for exercising and barriers to exercising. Each meaning unit was condensed into codes describing the same phenomena and grouped into subcategories that finally formed categories on the manifest level. Finally, to achieve a consensus of the analysis, discussions and reflections of the text took place between the main author and the co-authors. Graneheim and Lundman [35] emphasise the importance of moving back and forth between the whole and parts of the text during the analysis to increase validity, which was considered throughout the process.

### 2.5. Ethical Considerations

The Regional Ethical Review Board at Lund University, Sweden, approved the study (2017/98). The study was performed according to the four ethical principles of the Declaration of Helsinki 2013: information, consent, confidentiality, and utility [40,41]. Participants aged 16 and above or the parents of younger participants consented in writing, and all were informed that they could withdraw from the study at any time [40]. After the interviews, all participants were allowed to discuss any thoughts or emotions that had emerged with healthcare professionals at CAP.

## 3. Results

The results revealed adolescents´ experiences of facilitators for and barriers to maintaining exercise after a group-based intervention for depression. Facilitators that emerged for maintaining exercise were improved self-esteem and a supportive environment, while barriers to maintaining exercise were disease burden and lack of a supportive environment (Table 2).

### 3.1. Facilitators for Maintaining Exercise

Adolescents´ experiences of facilitators for maintaining exercise after participating in an exercise intervention for depression, emerged in two categories: greater self-esteem and a supportive environment.

#### 3.1.1. Greater Self-Esteem

This category was based on the subcategories: companionship while exercising and achievement of exercise results. The subcategories described that exercising together with someone and achieving exercise results, such as improved physical and mental health, created greater self-esteem and facilitated maintenance of exercise after the intervention.

##### Companionship While Exercising

The adolescents described it as more manageable, fun, and safer to exercise with someone they already knew or got to know during the intervention. Being comfortable and exercising together with someone being at the same exercise level and understands what it is like to be depressed boosted their self-esteem and motivated continuing to exercise. Adolescents felt a responsibility to exercise when exercising together with a family member, friend, or an exercise group, even though they at times were reluctant to join the session.
“Yes, or well now when I exercise than I’m expected to come to the exercise session. And sometimes it’s been that I’ve felt that l really don’t have the energy to do it today, so than I’ve felt, that they expect me to come. And when they ask "where are you” it doesn’t feel good to say "I couldn’t manage to come today", or it feels difficult. So it’s like a bit of encouragement to get me going, and afterwards, I feel that if I´d stayed at home it would have been a mistake.”(Adolescent no. 4) 

##### Achievement of Exercise Results

The adolescents described different effects from their exercising, such as greater strength, aerobic capacity, and enhanced wellbeing, e.g., feeling happier, more curious, and less depressed. These experiences boosted their self-esteem and motivated them to both continue exercising and to be more physically active after the intervention.
“Yes at *** I did feel a great difference from before I stared until it ended. I was much, more alert. I liked myself more afterwards.”(Adolescent no. 6) 

The effects from the exercise intervention boosted their self-esteem when they began to trust their ability to achieve more than they had done before. Exercising to increase wellbeing or reach exercise results were often dependent on the adolescents’ knowledge about why exercise is healthy, based on both their own experiences and external influences. Together these experiences generated greater self-esteem among the adolescents and facilitated continued exercise after the intervention.
“It’s for your self-esteem and self-confidence that you have the energy to do it. Especially after you feel the difference from how you felt before. That you have the energy to do lots of things.” (Adolescent no. 10) 

#### 3.1.2. A Supportive Environment

This category was based on the subcategories: *accessibility to exercise,* and *exercise coaching*. These aimed to describe the importance of the social and geographical environment around the adolescents, which influenced the maintenance of exercise after the intervention.

##### Accessibility to Exercise

When the adolescents were able to take themselves by public transport or cycle to an activity, the short distance was spoken of as facilitating the maintenance of the exercise after the intervention. However, when the distance became an obstruction for them to participate in an activity, the accessibility by having someone to drive them to the activity (e.g., a family member) was said to be important. Having access to social support from family and friends facilitated engaging in exercise after the intervention, where they were able to attend by having support during the exercise or received help to maintain the exercise routine.
“And when l really didn’t want to do it, it was my parents who forced me. Because they knew that I’d feel better afterwards. It´s always been difficult for me to go and exercise or anything like that and they want me to do this.” (Adolescent no. 10)

##### Exercise Coaching

Avoiding long inactive periods by keeping an exercise routine, having an exercise goal, and scheduling exercise was required for maintaining regular exercise after the intervention. Adolescents who continued exercising with a family member, a personal trainer or other trainers after the intervention thought that coaching generated support for maintaining an exercise routine. Not having to plan the exercise session, in terms of what to do and for how long or hard, was preferable and made it possible to relax and focus on the performance and the exercise goal. Having someone to determine what to do generated structure for the exercise and simplified the achievement of the exercise goal. Knowing that the exercise was performed correctly created feelings of security.
“It had a really great effect in that I don’t have to arrange anything myself or think about continuing and preparing and things like that, instead, I go there and do what I’m told to do. It´s very good for me and makes it easier to get it done.” (Adolescent no. 4)

### 3.2. Barriers to Maintaining Exercise 

Adolescents´ experiences of barriers to maintaining exercise after conducting an exercise intervention for depression emerged in two categories: the disease burden, and lack of a supportive environment.

#### 3.2.1. The Disease Burden

This category was based on the subcategories: fatigue, social anxiety, and a lack of drive. These aimed to describe difficulties in maintaining exercise as well as being physically active when depression is present.

##### Fatigue

The adolescents talked about often feeling a desire to exercise but not having enough strength and energy. It was difficult to prioritise exercise or even be physically active due to feeling tired after school. The lack of strength and energy created a loss of interest and enthusiasm, which hindered taking the initiative to continue exercise after the intervention.
“I´d really like to start exercising more, and I’m really looking forward to it, it’s just the lack of energy now.” (Adolescent no. 5)

##### Social Anxiety

Interacting and cooperating with new peers in an exercise group or at a gym was difficult and could create resistance to continuing exercise due to feeling uncomfortable. Similarly, the adolescents experienced discomfort when exercising with peers who did not know what it is like to be depressed. Feelings of being observed created a fear of making mistakes, which occurred when unfamiliar people were in an exercise group.
“It’s easy to flee from it. And I thought it was a bit difficult at the gym when there were people there because it was not something I was used to, exercise with a lot of unknown people like that. It created a little resistance to going there and then it was easy to miss a day.” (Adolescent no. 6)

##### A Lack of Drive

Missing an exercise routine often entailed a postponed exercise and hindered continued exercising. Exercising was described as strenuous, challenging, and time-consuming, due to constant muscle soreness and tiredness, while school and friends were a priority and perceived as more important. The weather was also described as influencing the willingness to engage in activities. A lack of exercise effect or perceptions that exercise did not increase wellbeing influenced motivation to continue exercising or being physically active. Some of those who had experienced wellbeing described that they no longer had a drive or a reason to continue exercising or be physically active after the exercise intervention.
“Is it something else that creates resistance, that I don’t want to or can’t exercise now?”(Interviewer)
“Something that’s to do with school, such as homework or schoolwork. You feel that it’s more important because it’s about school.” (Adolescent no. 9)

#### 3.2.2. Lack of a Supportive Environment

This category was based on the subcategories: a lack of social support, and a lack of structural support. These aimed to describe difficulties in maintaining exercise when not having a supportive environment

##### A Lack of Social Support

Not having social support to continue exercise after the intervention from either family members, a personal trainer or other trainers was spoken of as being difficult by the adolescents. It was difficult to stay motivated, to structure the exercise, and reach the intense exercise level unaccompanied without this support. Low intense physical activities, such as walking, were preferable instead. Exercises were easily skipped and not prioritised when there were no demands or expectations. There was also, however, a risk of exercising too strenuously when exercising alone, resulting in dizziness or blurred vision.
“Yes, it was, it was a little more difficult, because I went to the gym and exercised alone and it was harder to press myself as much and it was easier to just miss one and think that I´ll do it tomorrow.” (Adolescent no. 4) 

##### A Lack of Structural Support

Limited economic support was mentioned as a barrier to maintaining exercise after the intervention by the adolescents. The need for sportswear or other workout equipment could prevent them from engaging in exercise and not being able to afford organised sports activities. Having to travel a long distance affected their ability to perform when they were dependent on public transport. Furthermore, age restrictions at fitness centres were mentioned as a barrier to maintaining exercise, where adolescents felt they were neglected. Physical Education (PE) was not scheduled for everyone in school, which thus negatively influenced adolescents with less PE.
“I’ve always loved riding a horse since I was a child, but that also costs a lot of money. So that’s not possible.” (Adolescent no. 1) 
“Have you exercised afterwards?” (Interviewer) 
“Yes, a little now and then. I still had PE at school after the intervention, so then I had it twice a week. Then it became less because the PE at school ended when we started the second year.” (Adolescent no. 2) 

## 4. Discussion

This study highlights the importance of boosting self-esteem among adolescents with depression by providing companionship during exercise and achieving exercise results, and creating a supportive environment regarding accessibility to exercise and exercise coaching to facilitate maintenance of exercise. Furthermore, a lack of a supportive environment in terms of lack of social support and a lack of structural support as well as the disease burden of depression due to fatigue, social anxiety, and a lack of drive were barriers to maintaining exercise.

Enhanced self-esteem facilitated maintenance of exercise among the adolescents. Self-esteem was boosted when experiencing companionship, exercising together with someone who understood what it is like to be depressed. Exercise results generated a sense of achievement which also boosted self-esteem, and adolescents began to trust their ability to achieve more than they had done previously. Self-esteem and depression are linked among adolescents, and when they experience low self-esteem they seek to avoid potentially harmful experiences, such as not engaging in exercises to protect themselves from further harm [25,42,43]. Evidence suggests that interventions aiming to enhance self-esteem might be useful for reducing depression [43]. Our findings support that interventions aiming to enhance self-esteem, like patient group training and increased fitness, facilitate the maintenance of exercise among adolescents with depression.

A supportive environment was found to facilitate exercise maintenance among the adolescents. Adolescents who experienced support from families, friends, personal trainers, or other trainers to continue exercising spoke of coaching as a facilitator for maintaining exercise after the intervention. The support increased the motivation to maintain exercise, which is in line with earlier findings from healthy adolescents [28,30] and adults with depression [44]. Providing support during an exercise intervention by healthcare professionals and personal trainers is required for the participators to attain adherence [45]. Social support may help adolescents and adults with depression to overcome psychological barriers, such as low self-esteem and lack of motivation towards exercise [32,33,34,46]. These findings are in line with the results of the study, that a supportive environment around the adolescents is of importance to maintain exercise after an intervention. 

The disease burden of depression hindered adolescents from maintaining exercise after the intervention, due to feelings of fatigue, social anxiety, and a lack of drive. Even though adolescents were motivated to continue exercising after the intervention, a lack of energy was experienced as a hinder to exercising. Previous research has shown that young adults with depressive symptoms are less likely to engage in exercise, and the inactivity may be dependent on the inactivity of friends and family [25]. This illustrates the difficulty of exercising when depression is present and the importance of creating a safe environment with continuous support, to facilitate maintenance of exercise. 

Lack of a supportive environment hindered adolescents with depression from maintaining exercise after the intervention in the present study. Exercise was easily neglected and not prioritised among adolescents without a supportive environment. Missing structural support for exercising, such as a lack of workout equipment or sportswear, limited accessibility to a fitness centre, and limited economic support prevented the adolescents from engaging in exercise. These results are in line with findings by Fraser et al., where limited economy was found to be a barrier to engaging in exercise among adults with depression [47]. It has been suggested that schools could reduce the economic barrier to leisure activities by offering longer breaks to encourage physical activities or more PE lessons among healthy adolescents [32]. The school environment restricted exercise among adolescents with depression in the present study, since they felt uncomfortable exercising with peers who did not understand what it is like to be depressed. Research on experiences from adolescents with depression reveal beliefs among their peers that depression does not exist [48]. Increased perceived safety in schools among adolescents with depression could be created with social support from families and teachers [49]. Greater social support may be important when encouraging adolescents with depression to maintain exercise in a school setting. Our results add to the suggestions of various supportive mechanisms like economic support, access to a safe venue and nudging from family and teachers to be considered in order to facilitate exercise maintenance after an exercise intervention.

### 4.1. Strengths and Limitations

A strength of the study was that data from the 14 interviews attained an adequate number of meaning units to detect variations and similarities, supporting saturation. Trustworthiness in qualitative studies is attained by the richness of the data and not by sample size [50]. The text from the interviews was deemed rich on account of the 440 meaning units that emerged. Facilitating the readers’ ability to distinguish the common thread throughout the research enhances credibility [36]. The analysis process comprised systematic handling of the data, thus facilitating confirmability, where the main author read all the transcribed text and then discussed the findings with the co-authors to achieve a consensus in the analysis. The findings were illustrated with appropriate quotations in order to enhance the confirmability of the result. All the participants were interviewed individually, where the same interview questions were asked, thus strengthening dependability [35]. There may be difficulties interviewing adolescents with depression due to their vulnerabilities [51]; participants may, for example, have answered questions in a certain way to avoid disclosure. The interviews were performed by an experienced nursing researcher (IL) familiar with daily life challenges in adolescence. This enabled the interviewer to engage with the participants and build trust [52,53]. Recruiting participants from CAP strengthens the result´s transferability within the patient group of adolescents with depression [35]. However, a limitation to the transferability was recruiting participants from the same clinic with a rather homogenous ethnic and cultural background. Further, participants were heavily burdened by the long duration of depression and significant comorbidities, which might have exposed them to greater difficulties to maintaining exercise than patients with primary-level depression, thus magnifying barriers and facilitators. 

### 4.2. Implications

Healthcare professionals needs to include interventions into their practice to enhance continuous support in order to facilitate exercise maintenance after an exercise intervention among adolescents with depression. These interventions could amount to liasing with schools, informing families about their very crucial support and their role to find, and financially and practically support access to, an acceptable venue to exercise. Future research should explore which strategies are the most effective and how these interventions can be incorporated into usual care for depression in adolescence. 

## 5. Conclusions

Greater self-esteem and a supportive environment for exercising can facilitate the maintenance of exercise among adolescents with depression. The disease burden of depression in terms of fatigue, social anxiety, and lack of drive is a pronounced barrier to maintaining exercise for this target group. These findings suggest that adolescents with depression need a continuous and tailored effort to enhance self-esteem and support in order to maintain exercise after an exercise intervention. 

## Figures and Tables

**Table 1 ijerph-18-05427-t001:** Participant characteristics at the one-year follow-up.

Participants (*n* = 14)	Median (Min-Max)
Age (years)	17 (14–19)
Body mass index (kg/m^2^)	28.6 (18.7–37.3)
Depression score QIDS-Clinician rated *	4 (1–9)
Depression score QIDS-Self-reported *	6 (1–18)

*****
Quick Inventory of Depressive Symptomatology—Adolescent Version (Total scores 0–27).
Mild depression 6-10 score, moderate depression 11–15 score, severe depression 16–20 score and very severe depression above 21 scores.

**Table 2 ijerph-18-05427-t002:** Content areas, categories, and sub-categories exploring adolescents´ experiences of facilitators for and barriers to maintaining exercise after a group-based intervention for depression.

Content Areas	Categories	Sub-Categories
Facilitators for maintaining exercise	Greater self-esteem	Companionship while exercising
		Achievement of exercise results
	A supportive environment	Accessibility to exercise
		Exercise coaching
Barriers to maintaining exercise	The disease burden	Fatigue
		Social anxiety
		A lack of drive
	Lack of a supportive environment	A lack of social support
		A lack of structural support

## Data Availability

Not applicable. The data will not be shared as ethics approval for the study requires that the transcribed interviews are kept in locked files, accessible only to the researchers.

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
