# Peer review of "Adolescents’ Experiences of Facilitators for and Barriers to Maintaining Exercise 12 Months after a Group-Based Intervention for Depression"

_ijerph, 2021, doi:10.3390/ijerph18105427_

Round 1

Reviewer 1 Report

Overall value of article:  Important topic (exercise in adolescent depression), important focus (what helps and what gets in the way of adolescents continuing exercise after a training session).

Methodology:  very good, well explained.

Clarity of writing:  good.

Quality of English: overall good, with one important exception: the authors use the word "facilitator" to refer to factors that increased the likelihood of the adolescents' maintaining their exercise practice.  For me - you can check with others - "facilitator" refers to a person who facilitates, not an impersonal thing.  Others might disagree, but I would suggest that the authors replace "facilitator" with "facilitating factor" or something similar.

In addition, there is one spelling error - at end of first paragraph on page 2, should be 'studies are few' not studies is few.

Author Response

Response to reviewers

We thank the editor and reviewers for a thorough review and helpful comments to improve the manuscript.

Please find the reviewers’ original comments and our response below (the latter using italics). In the revised manuscript, changes have been marked by colour to identify corrections.

Reviewer: 1

Comments to the Author

Overall value of article:  Important topic (exercise in adolescent depression), important focus (what helps and what gets in the way of adolescents continuing exercise after a training session).

Methodology:  very good, well explained.

Clarity of writing:  good.

Answer: Thank you

Quality of English: overall good, with one important exception: the authors use the word "facilitator" to refer to factors that increased the likelihood of the adolescents' maintaining their exercise practice.  For me - you can check with others - "facilitator" refers to a person who facilitates, not an impersonal thing.  Others might disagree, but I would suggest that the authors replace "facilitator" with "facilitating factor" or something similar.

Answer: Thank you, we have taken this suggestion into account. However, we can not find qualitative studies on Pubmed using the term facilitating factor and therefore we choose to use facilitator instead of facilitating factor. Also, the text immediately following the “facilitator” explains that the term refers to a factor and not to a person so confusion on this part should be avoided.

In addition, there is one spelling error - at end of first paragraph on page 2, should be 'studies are few' not studies is few.

Answer: Thank you for your observation. We have changed this

Reviewer 2 Report

General

Authors explore in their manuscript ‘Adolescents’ experiences of facilitators for and barriers to maintaining exercise 12 months after a group-based intervention for depression’ the facilitators and barriers adolescents experience after a 12 months of a group-based intervention consisting of exercise for depression. Longitudinal mental health in adolescents is understudied, however some minor work should be done before this manuscript can be published.

Title

-

Names

Couldn't you write them differently, with the Orcid as a sign?

Abstract

Background

-

Methods

Add (after the sentence with: A qualitative content analysis was used.) a sentence announcing how you will report the data, like:

First, facilitators for maintaining exercise were invented; finally the barriers.

Results

Why is <Facilitators> bolded?

Conclusion

Start with an answer to the aim.

Key words

Give them a different sequence, not <adolescents; barriers; depression; exercise; facilitators> but <adolescents; facilitators; barriers; depression; exercise>

Introduction

I believe with you that <Knowledge of these factors can help the clinic to tailor the treatment.> My question then would be, why did you choose for such a low N? You will not reach a sufficient knowledge of these factors?

Methods

Sample

Measures

Statistical analyses / Reporting

Please rewrite this section: First, we … . Then, we … . Next, we … . Finally, we … . The readership easier grabs what you did and in which order the Results will be shown.

Results

Please make of <the intervention> <the intervention.>

Could you add to <(Adolescent no. 4)> the gender and age? Like: (M Adol 4; 14 yrs) – in some of the adolescents gender or age is more important than the more or less number by chance you gave them.

Italicize <And when l really didn’t want to do it, it was my parents who forced me. Because they knew that I’d feel better afterwards. It´s always been difficult for me to go and exercise or anything like that and they want me to do this. (Adolescent no. 10)>.

Italicize <It had a really great effect in that I don’t have to arrange anything myself or think about continuing and preparing and things like that, instead, I go there and do what I’m told to do. It´s very good for me and makes it easier to get it done. (Adolescent no. 4)>

Discussion

Please keep in mind the following structure for writing a Discussion:

para1                      start with repeating the research question + answer this without any comments or interpretation – you did this.

Para2,3,#               start a new para, 1 topic per para, and start this para with one of your findings – which then defines the content of the para. Relate your finding to earlier published references; in the last sentence show how existing knowledge changed as a result of your finding.

Strengths and limitations (you mention this Methodological discussion)

Mention strengths! Limitations are those issues which might bias your findings (are there sources of bias); please add the direction of the findings.

Implications

(split into: for practice/policy, for future research)

Add such a para.

Conclusion

Please rewrite this Discussion part

Tables, Figures

References

Author Response

Response to reviewers

We thank the editor and reviewers for a thorough review and helpful comments to improve the manuscript.

Please find the reviewers’ original comments and our response below (the latter using italics). In the revised manuscript, changes have been marked by colour to identify corrections

Reviewer 2

Comments and Suggestions for Authors

General

Authors explore in their manuscript ‘Adolescents’ experiences of facilitators for and barriers to maintaining exercise 12 months after a group-based intervention for depression’ the facilitators and barriers adolescents experience after a 12 months of a group-based intervention consisting of exercise for depression. Longitudinal mental health in adolescents is understudied, however some minor work should be done before this manuscript can be published.

Answer: Thank you.

Names

Couldn't you write them differently, with the Orcid as a sign?

Answer: Thank you for your observation. We have moved the Orcids to the end of the manuscript with a separate headline.

Abstract

Methods

Add (after the sentence with: A qualitative content analysis was used.) a sentence announcing how you will report the data, like:

First, facilitators for maintaining exercise were invented; finally the barriers.

Answer: Thank you. We have clarified the method in the abstract.

Results

Why is <Facilitators> bolded?

Answer: Thank you for your observation.

Conclusion

Start with an answer to the aim. 

Answer: Thank you. We have clarified the conclusion by adding “In conclusion” and “The findings of this study”.

Key words

Give them a different sequence, not <adolescents; barriers; depression; exercise; facilitators> but <adolescents; facilitators; barriers; depression; exercise>

Answer: Thank you. We have changed the sequence of the keywords into the following: adolescents; facilitators; barriers; depression; exercise.

Introduction

I believe with you that <Knowledge of these factors can help the clinic to tailor the treatment.> My question then would be, why did you choose for such a low N? You will not reach a sufficient knowledge of these factors?

 Answer: Thank you for your question. In qualitative studies is trustworthiness attained by the richness of the data and not by sample size (1,2). We outline this position at the beginning of the headline Strengths and limitations.

1) Sandelowski, M. Sample size in qualitative research. Research in nursing & health 1995, 18, 179-183.

2) Malterud, K., Siersma, V. D., & Guassora, A. D. (2016). Sample size in qualitative interview studies: guided by information power. Qualitative health research, 26(13), 1753-1760.

Methods 

Statistical analyses / Reporting

Please rewrite this section: First, we … . Then, we … . Next, we … . Finally, we … . The readership easier grabs what you did and in which order the Results will be shown.

 Answer: Thank you for your suggestion. We have updated the section Data analysis in order to facilitate the readership.

Results

Please make of <the intervention> <the intervention.>

Answer: Thank you for your observation. We have added the full stop.

Could you add to <(Adolescent no. 4)> the gender and age? Like: (M Adol 4; 14 yrs) – in some of the adolescents' gender or age is more important than the more or less number by chance you gave them.

Italicize <And when l really didn’t want to do it, it was my parents who forced me. Because they knew that I’d feel better afterwards. It´s always been difficult for me to go and exercise or anything like that and they want me to do this. (Adolescent no. 10)>.

Italicize <It had a really great effect in that I don’t have to arrange anything myself or think about continuing and preparing and things like that, instead, I go there and do what I’m told to do. It´s very good for me and makes it easier to get it done. (Adolescent no. 4)>

 Answer: Thank you for your question. This is a good proposal to enhance the understanding of the narratives of the adolescents. However, to protect the identity of the adolescents and to maintain complete confidentiality through the study, we have considered not publish the age or the gender due to the sample size with only four male participants.

Discussion

Please keep in mind the following structure for writing a Discussion:

para1                      start with repeating the research question + answer this without any comments or interpretation – you did this.

Para2,3,#               start a new para, 1 topic per para, and start this para with one of your findings – which then defines the content of the para. Relate your finding to earlier published references; in the last sentence show how existing knowledge changed as a result of your finding.

 Answer: Thank you very much for this helpful suggestion. We have clarified the findings in chronological order at the beginning of every paragraph. We have also clarified how existing knowledge has changed as a result of our findings at the end of every paragraph. This way, the discussion has become more clear for the reader.

Strengths and limitations (you mention this Methodological discussion)

Mention strengths! Limitations are those issues which might bias your findings (are there sources of bias); please add the direction of the findings.

 Answer: Thank you. We have updated the following section to Strengths and limitations and clarified the direction of the findings.

Implications

(split into: for practice/policy, for future research)

Add such a para.

 Answer: Thank you. We have added a new section of Implication.

Please rewrite this Discussion part

Thank you. We have rewritten the discussion part as you suggested. We have also clarified how existing knowledge has changed as a result of our findings at the end of every paragraph and added a new section of Implication.